# A CENH3 mutation promotes meiotic exit and restores fertility in SMG7-deficient Arabidopsis

**Claudio Capitao**[1], **Sorin Tanasa**[2,3], **Jaroslav Fulnecek**[2], **Vivek K. Raxwal**[2], **Svetlana Akimcheva**[1], **Petra Bulankova**[1], **Pavlina Mikulkova**[2], **Lucie Crhak Khaitova**[2], **Manikandan Kalidass**[4], **Inna Lermontova**[4], **Ortrun Mittelsten Scheid**[1], **Karel Riha**[2]*

**1** Gregor Mendel Institute, Austrian Academy of Sciences, Vienna, Austria, **2** Central European Institute of Technology (CEITEC), Masaryk University, Brno, Czech Republic, **3** National Centre for Biomolecular Research, Faculty of Science, Masaryk University, Brno, Czech Republic, **4** Leibniz Institute of Plant Genetics and Crop Plant Research, Gatersleben, Germany

* karel.riha@ceitec.muni.cz

**Data Availability Statement:** All relevant data are within the manuscript and its Supporting information files.

## Abstract

Meiosis in angiosperm plants is followed by mitotic divisions to form multicellular haploid gametophytes. Termination of meiosis and transition to gametophytic development is, in Arabidopsis, governed by a dedicated mechanism that involves SMG7 and TDM1 proteins. Mutants carrying the *smg7-6* allele are semi-fertile due to reduced pollen production. We found that instead of forming tetrads, *smg7-6* pollen mother cells undergo multiple rounds of chromosome condensation and spindle assembly at the end of meiosis, resembling aberrant attempts to undergo additional meiotic divisions. A suppressor screen uncovered a mutation in centromeric histone H3 (CENH3) that increased fertility and promoted meiotic exit in *smg7-6* plants. The mutation led to inefficient splicing of the CENH3 mRNA and a substantial decrease of CENH3, resulting in smaller centromeres. The reduced level of CENH3 delayed formation of the mitotic spindle but did not have an apparent effect on plant growth and development. We suggest that impaired spindle re-assembly at the end of meiosis limits aberrant divisions in *smg7-6* plants and promotes formation of tetrads and viable pollen. Furthermore, the mutant with reduced level of CENH3 was very inefficient haploid inducer indicating that differences in centromere size is not the key determinant of centromere-mediated genome elimination.

## Author summary

Meiosis is a reductional cell division that halves number of chromosomes during two successive rounds of chromosome segregation without intervening DNA replication. Such mode of chromosome segregation requires extensive reprogramming of the cell division machinery at the entry to meiosis, and inactivation of the meiotic program upon the formation of haploid spores. Here we showed that Arabidopsis partially deficient in the RNA decay factor SMG7 fail to exit meiosis and continue with attempts to undergo additional cycles of post-meiotic chromosome segregations without genome replication. This results

**Funding:** This work was supported from the European Regional Development Fund-Project 'REMAP' (No. CZ.02.1.01/0.0/0.0/15_003/0000479 to K.R.), Doctoral School "Chromosome Dynamics" of the Austrian Science Fund (FWF W1238 to K.R. and O.M.S.), Vienna Science and Technology Fund (WWTF LS13-057 to O.M.S.) and the German Federal Ministry of Education and Research (Plant 2030, Project 031B0192NN, HaploTools, to I.L.). The core facility CELLIM of CEITEC is supported by MEYS CR (LM2018129 Czech-BioImaging). The funders had no role in study design, data collection and analysis, decision to publish, or preparation of the manuscript. Funders web sites: MEYS: https://www.msmt.cz/ Austrian Science Fund: https://www.fwf.ac.at/ Vienna Science and Technology Fund: https://www.wwtf.at/ German Federal Ministry of Education and Research https://www.bmbf.de/.

**Competing interests:** The authors have declared that no competing interests exist.

in a reduced number of viable pollen and diminished fertility. To find genes involved in meiotic exit, we performed a suppressor screen for the SMG7-deicient plants that re-gain fertility. We found that reducing the amount of centromeric histone partially restores pollen formation and fertility in *smg7* mutants. This is likely due to inefficient formation of centromere-microtubule interactions that impairs spindle reassembly and re-entry into aberrant rounds of post-meiotic chromosome segregation.

## Introduction

A sexual life cycle consisting of alternating haploid and diploid life forms is the defining feature of eukaryotes. Entry into the haploid phase requires meiosis, a reductional cell division that forms four haploid cells from a single diploid precursor. It involves segregation of homologous chromosomes in the first meiotic division that is followed, without intervening DNA replication, by segregation of sister chromatids in the second meiotic division. In contrast to the mitotic cell division machinery, meiosis requires mechanisms for tethering homologous chromosomes via recombination in prophase I, sister kinetochore monoorientation and protection of centromeric cohesion in metaphase-anaphase I, and inhibition of DNA replication in interkinesis [1,2]. While the sequence of meiotic events is evolutionarily highly conserved, regulation of meiosis and its position in the context of the life cycle differ across diverse phylogenetic units [3,4].

Meiosis in angiosperm plants occurs in megaspore and pollen mother cells located in pistils and anthers, respectively, and leads to the formation of haploid spores. Rudimentary multicellular gametophytes carrying male and female gametes are formed by subsequent mitotic divisions. A number of genes involved in induction, progression, and termination of the meiotic program have been identified in plants. Genes required for meiotic fate acquisition and progression through early meiotic events include the transcription factor *SPOROCYTELESS*, RNA binding protein *MEL2*, AMEIOTIC1/SWITCH1, and *RETINOBLASTOMA RELATED1* [5–9]. Redox status and small-RNA-mediated gene silencing have also been implicated in establishing meiotic cell fate [10–12]. Entry into meiosis is further accompanied by the induction of genes required for core meiotic functions [13,14].

Progression through the meiotic cell cycle is driven by cyclin dependent kinases (CDKs), in Arabidopsis mainly by CDKA;1, the key CDK that is also required for mitosis [15]. CDKA;1 plays a role in regulating meiotic spindle organization, cytokinesis, as well as recombination and chromosome pairing [16–18]. CDKA;1 activity is modulated by several mechanisms to implement the meiotic program. CDK substrate specificity is determined through association with different cyclins. Several A- and B- type cyclins are expressed in Arabidopsis pollen mother cells (PMCs) [19] and CDKA;1 was found to interact with at least three of them. SOLO DANCERS (SDS) is a meiosis-specific cyclin that mediates phosphorylation of the chromosome axis assembly factor ASYNAPTIC 1 (ASY1) and is essential for homologous recombination and pairing [16,20]. A-type cyclin TARDY ASYNCHRONOUS MEIOSIS (TAM) and CYCB3;1 have been implicated in organization of the meiotic spindle and regulation of cell wall formation [16,19,21,22]. An important aspect of meiosis is the absence of S-phase in interkinesis between meiosis I and II. In yeasts and mammals, this is achieved via partial inhibition of the anaphase promoting complex (APC/C) after anaphase I, which results in residual CDK activity in interkinesis to prevent DNA replication. A similar mechanism was also suggested in Arabidopsis, where inactivation of the APC/C inhibitor OMISSION OF SECOND DIVISION 1 (OSD1) leads to premature meiotic exit after meiosis I [23,24].

Two genes have been implicated in terminating the meiotic program and enabling the transition to gametophytic development in Arabidopsis. *THREE DIVISION MUTANT1 (TDM1)/ MS5/POLLENLESS3* is a plant-specific gene that is exclusively expressed in meiocytes, and loss of its function results in male sterility [25–28]. Mutant PMCs fail to exit meiosis and the chromosomes of the haploid nuclei re-condense, nucleate four spindles, and attempt to undergo a third division [25,29]. Another gene required to terminate meiosis is *SUPPRESSOR WITH MORPHOGENETIC EFFECTS ON GENITALIA7 (SMG7)*, an evolutionary conserved protein involved in nonsense-mediated RNA decay (NMD). Arabidopsis *smg7*-null mutants are NMD-deficient, exhibit stunted growth due to an upregulated immune response, and are infertile [30,31]. The infertility is caused by meiotic arrest in anaphase II and inability to exit meiosis. Analysis of plants with a hypomorphic *smg7-6* allele that contains a T-DNA insertion in the diverged C-terminal domain of the gene indicate that the meiotic function of SMG7 is not connected to its role in NMD [32], but the mechanism of its action in meiosis remains unknown.

We reasoned that mutations that restore fertility and increase pollen count in *smg7-6* mutants might help identify genes that affect meiotic exit in Arabidopsis and therefore performed a genetic suppressor screen in this background. We identified two suppressor lines with a mutation in the *CENH3* gene, which encodes the centromeric variant of histone H3. This mutation does not alter the amino acid sequence of the protein but leads to inefficient splicing of *CENH3* mRNA and a substantial reduction of CENH3 protein levels. We describe the consequences of decreased CENH3 for meiosis, mitosis, and centromere-mediated induction of haploid plants.

## Results

### Meiosis in *smg7-6* PMCs is followed by multiple rounds of spindle reassembly prior to polyad formation

The multiple functions of Arabidopsis SMG7 were described in our previous studies of an allelic series of T-DNA insertion mutants in this gene [32]. The *smg7-1* and *smg7-3* alleles, disrupted in the conserved central domain, are NMD-deficient and exhibit retarded growth and defects in leaf development and shoot branching (S1 Fig). In addition, *smg7-1* and *smg7-3* mutants are completely sterile (Fig 1A)[30]. In contrast, *smg7-6* mutants, which contain a T-DNA insertion downstream of the conserved central domain, grow normally and are only mildly deficient in NMD [31](S1 Fig). Nevertheless, fertility is strongly reduced in early flowers of *smg7-6* plants, but, in contrast to *smg7-1* and *smg7-3*, some seeds are produced from late flowers. Anthers of these flowers have approximately 10-times less viable pollen than wild type (Fig 1A). We noticed that seed production is approximately halved upon pollination of *smg7-6* mutants with wild type pollen indicating that the female gametophyte is also affected (S2 Fig). Because the *smg7-6* phenotype differed from the one detected in *smg7-1*, we transformed the mutant with the wild-type SMG7 gene construct. This complementation restored pollen production and confirmed that the reduced fertility is indeed caused by the *smg7-6* allele (S3 Fig).

Previous cytogenetic analysis performed on dissected *smg7-6* meiocytes showed PMCs seemingly arrested in anaphase II, as well as meiocytes that reached telophase II [32]. This observation can be interpreted as a reduced penetrance of the *smg7*-null phenotype in mutants with this allele, allowing occasional continuation into normal pollen development. If this were the case, one anther should contain a mixture of aberrant and normal meiocytes. To test this, we performed cytogenetic analysis of meiocytes in entire anthers. In contrast to our prediction, we observed anthers in which all meiocytes resembled aberrant anaphase II, as well as anthers with all meiocytes in telophase II (Fig 1B). This suggests that all meiocytes in *smg7-6* plants are

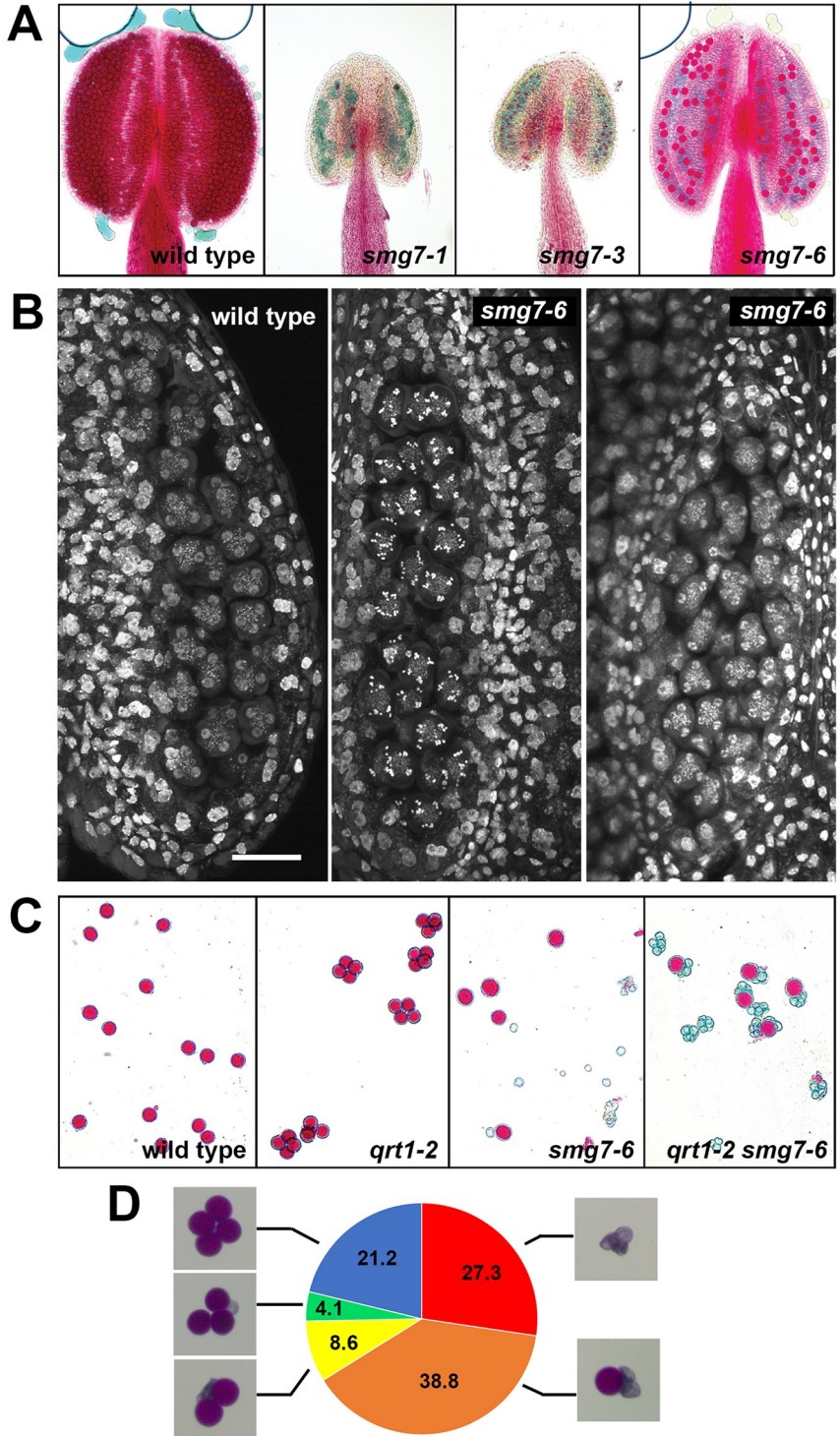

**Fig 1. Aborted pollen development in *smg7-6* mutants. (A)** Anthers of indicated mutants after Alexander staining. Viable pollen stain red. **(B)** Anther loculi in telophase II (Col-0, *smg7-6* left panel) and assumed aberrant anaphase II (*smg7-6*, middle panel) stained with DAPI. Scale bar corresponds to 20 μm. **(C)** Alexander staining of pollen produced by the indicated mutants. The *qrt1-2* mutation causes pollen from one tetrad to remain together. **(D)** Percentage of tetrads with one, two, three, four or no viable pollen. 245 tetrads from *qrt1-2 smg7-6* plants were counted.

able to reach telophase II and that the residual fertility cannot be attributed to the reduced penetrance of the phenotype observed in *smg7-1* mutant. To analyze pollen derived from the same meiocyte, we took advantage of the Arabidopsis *qrt1* mutant that is deficient in post-meiotic pollen separation, resulting in pollen tetrads that remain tethered together. Analysis of *qrt1-2 smg7-6* double mutants showed that tetrads mainly consisted of one viable and three aborted pollen (Fig 1C and 1D), eliminating the possibility that viable pollen originated from a bypass of anaphase II arrest and continuation of normal pollen development for all four products.

We next performed immunocytogenetic analysis of dissected *smg7-6* meiocytes stained with an α-tubulin antibody. We detected meiocytes containing four separate spindles, each attached to approximately five chromatids (Fig 2A). This phenotype resembles *tdm1* mutants, where haploid nuclei re-enter a third division without preceding DNA replication (Fig 2A). Nevertheless, *tdm1* and *smg7-6* mutations have slightly different phenotypes. In contrast to *smg7-6* plants, *tdm1* mutants are infertile and do not form any pollen (S4A Fig). We have also noticed that chromatids in *tdm1* mutants condense only partially during the aberrant post-meiotic division, while they fully condense in *smg7-6* plants (Fig 2B). *tdm1* is epistatic to *smg7-6* as *tdm1 smg7-6* double mutants are infertile and exhibit partially condensed chromosomes during the third meiotic division (S4B Fig).

To gain more insights into meiotic progression in *smg7-6* plants, we performed time-lapse analysis of PMCs using fluorescently labelled tubulin [22,33]. In wild type, microtubules undergo two rounds of spindle formation in meiosis I and meiosis II, separated by spindle disassembly in interkinesis (Fig 3A and 3B; S1 Movie). By tracking spindle formation, we determined the duration of metaphase I to telophase II, interkinesis, and metaphase II to telophase II in wild type to be 46, 38, and 39 min, respectively, with cytokinesis ensuing 203 min after meiosis II spindle disassembly (average values from 5 meiocytes). PMCs in *smg7-6* mutants underwent up to four additional cycles of spindle assembly and disassembly beyond meiosis II (Fig 3A and 3C; S2 and S3 Movies). We also noticed that meiosis I and II in *smg7-6* plants were extended to an average of 66 and 67 min, respectively (n = 5 meiocytes; Fig 3B). While the interkinesis II that preceded meiosis III was relatively brief (32 min), interkinesis in subsequent cycles were substantially longer (Fig 3B).

These observations indicate that *smg7-6* plants are unable to complete meiosis by regular cytokinesis. Instead, after meiosis II, *smg7-6* PMCs undergo several cycles of spindle assembly/disassembly and chromatin condensation/decondensation, leading to an unequal distribution of chromatin and formation of polyads that only exceptionally form functional haploid microspores and pollen. We did not detect any aneuploids among 19 cytologically scored progeny of *smg7-6* self-pollinated plants indicating that the majority of viable pollen do not contain supernumerary chromosomes. This indicates that the viable pollen are not derived from stochastic clustering of chromosomes, but rather represent events where the original set of chromosomes remained associated throughout the cycles of chromatin condensation and decondensation.

## Suppressor screening results in mutations that recover fertility of *smg7-6* plants

SMG7 and TDM1 are important elements of a regulatory network that governs meiotic exit in Arabidopsis [29]. To uncover additional genes and molecular processes required for transition from meiosis to pollen differentiation, we performed a forward genetic screen for mutants that restore the fertility of *smg7-6* plants. The screen is based on a characteristic feature of *smg7-6* mutants: the first 20–25 flowers on the main inflorescence bolt are infertile and only later flowers give rise to seeds (Fig 4A and 4B). Interestingly, pollen production and ability to form seeds are reduced to a similar extend in all flowers, indicating that increased fertility in late

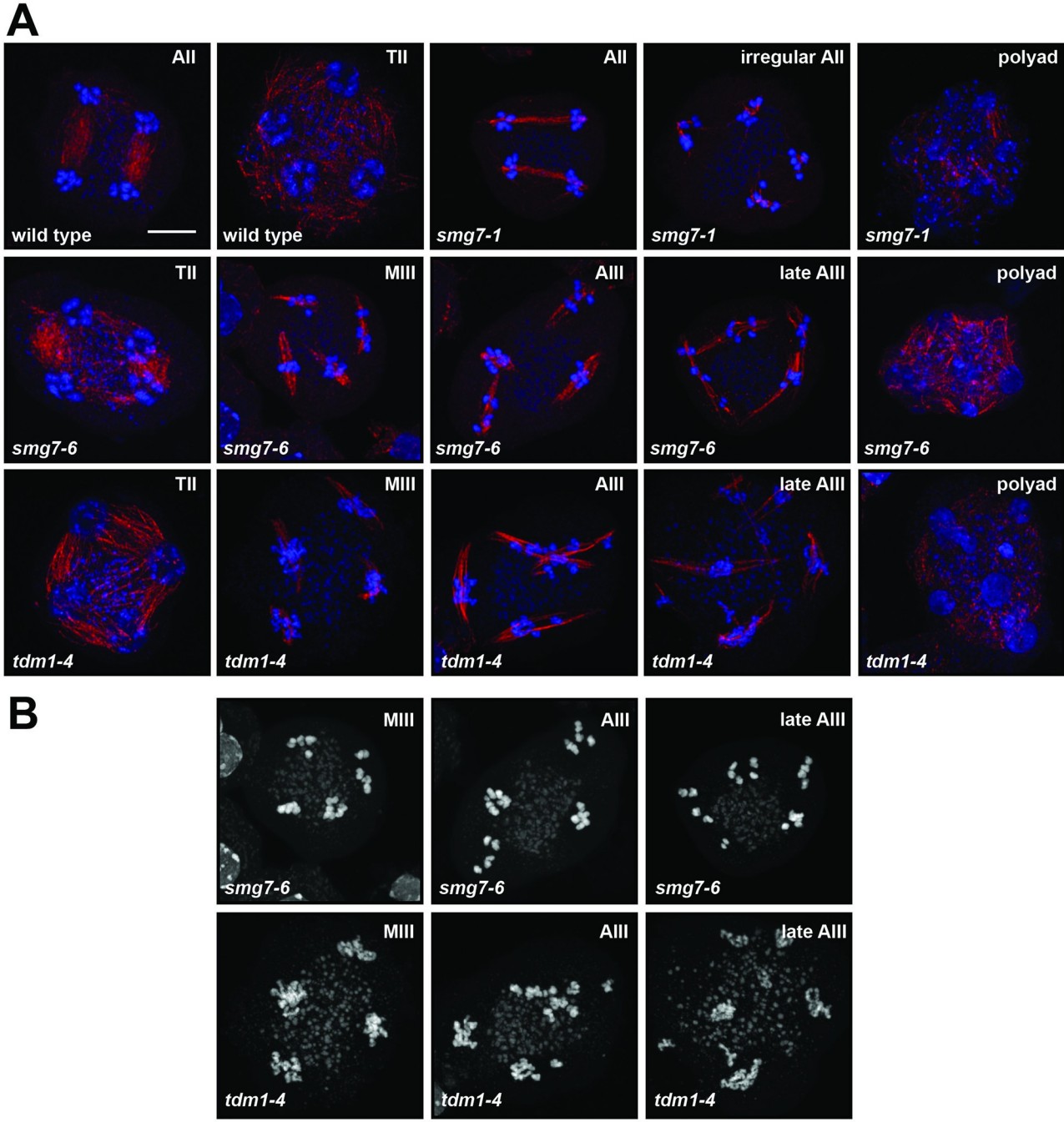

**Fig 2. Meiosis in *smg7* and *tdm1* mutants. (A)** Immunodetection of spindles using anti-α-tubulin antibody (red) in pollen mother cells. DNA is counterstained by DAPI (blue). AII—anaphase II, TII—telophase II, MIII—metaphase III, AIII—anaphase III. **(B)** Structure of chromosomes counterstained by DAPI in the third meiotic division.

flowers is not caused by changes in the meiotic program during plant aging, but rather by unknown physiological factors that affect pollination or fertilization (S2 Fig and Fig 4C and 4D). In the screen, we scored plants from the M2 population of ethyl-methanesulfonate (EMS)-mutagenized *smg7-6* plants and selected suppressor lines that produced seeds from the

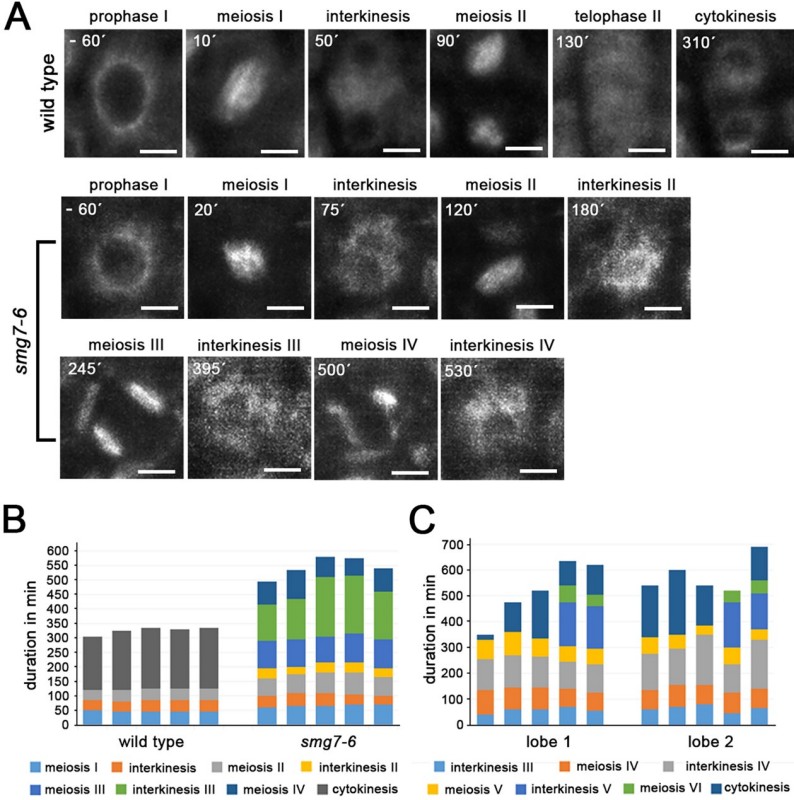

**Fig 3. Live imaging of meiotic progression in *smg7-6* PMCs. (A)** Time-lapse series of PMCs containing microtubules labeled with the p*RPS5A*::*TagRFP*:*TUB4* construct. Timepoints relate to nuclear envelope breakdown at the end of prophase I. Scale bars correspond to 5 μm. **(B)** Duration of indicated meiotic stages inferred from spindle dynamics in time-lapse movies of p*RPS5A*::*TagRFP*:*TUB4*. Five individual PMCs from each wild type and *smg7-6* plants are shown. **(C)** Duration of postmeiotic cycles of spindle assembly and disassembly in individual PMCs from two separate anther lobes of *smg7-6* mutants.

first 20 flowers on the main inflorescence bolt. Two suppressor lines, EMS30 and EMS155, are characterized in detail in this study.

Both lines produce seeds and substantially longer siliques from early flowers, although the siliques are still shorter than in wild type (Fig 4A and 4B). They also generate about 250 viable pollen per anther, which is six times more than *smg7-6* plants and approximately half of that in wild type (Fig 4C and 4D). While cytogenetic analysis revealed some meiocytes still undergoing a third meiotic division (Fig 5A), their abundance substantially decreased in comparison with *smg7-6* mutants (Fig 5B). Whole anther staining showed a number of meiocytes forming tetrads, while only a relatively small fraction exhibited the aberrant post-meiotic divisions (Fig 5C). Therefore, the aberrant exit from meiosis in *smg7-6* PMCs is at least partially suppressed in the double mutants, resulting in a relatively high level of correct tetrad formation and improved seed set.

## Restored fertility is caused by mutation in CENH3

To identify the mutations responsible for the restored fertility, we created mapping populations by backcrossing EMS30 and EMS155 lines to the parental *smg7-6*. The restored fertility phenotype segregated as a recessive trait. EMS-induced *de novo* mutations were identified by comparing genome sequencing data from pools of fertile B2 plants with the parental *smg7-6*

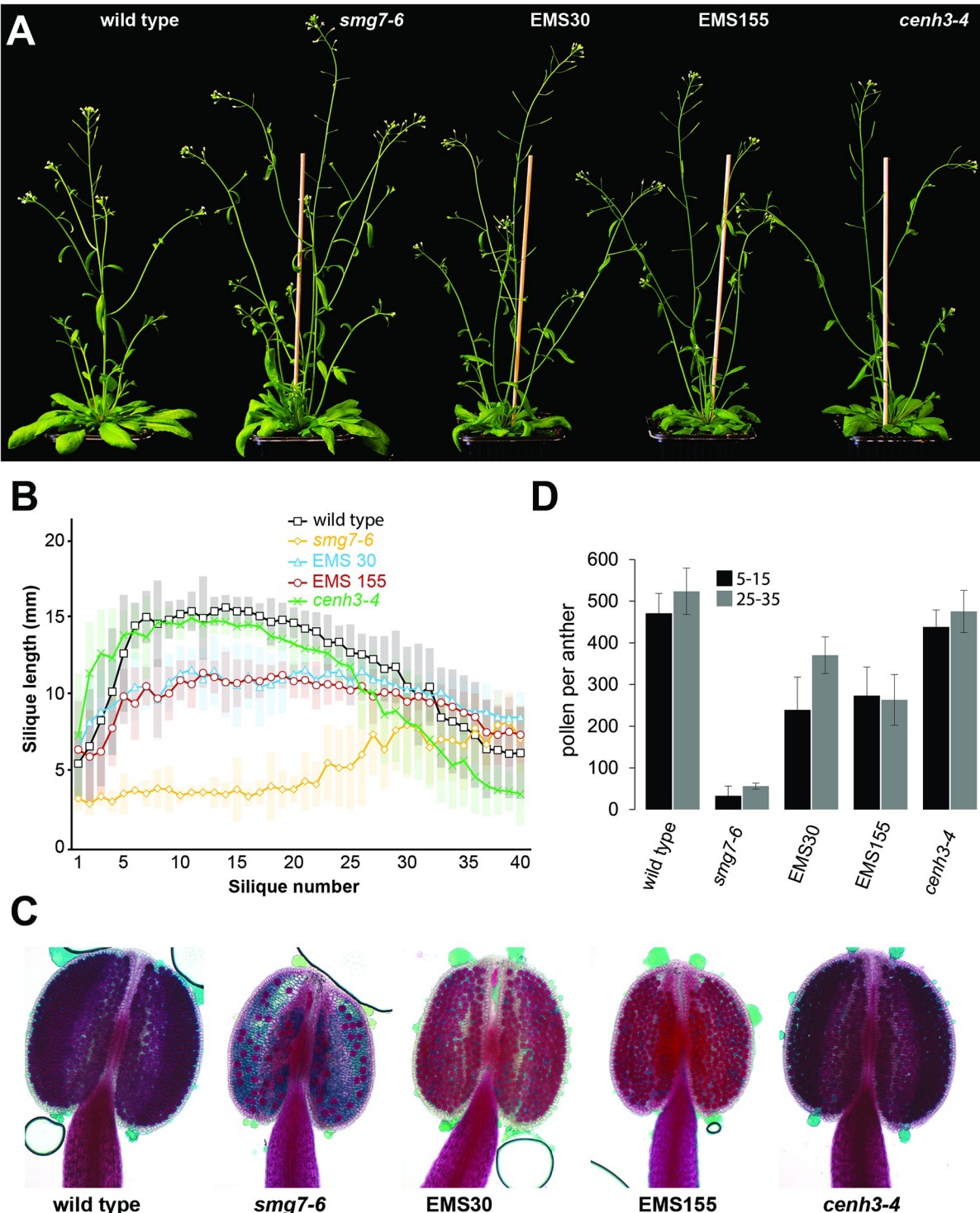

**Fig 4. Characterization of the EMS30 and EMS155 suppressor lines. (A)** Five week-old plants of indicated genotypes. **(B)** Analysis of silique length along the main inflorescence bolt in wild type (n = 16), *smg7-6* (n = 15), EMS30 (n = 16), EMS155 (n = 15) and *cenh3-4* (n = 16) plants. Position 1 corresponds to the oldest and position 40 to the youngest silique scored on the main bolt. Error bars represent standard deviations. **(C)** Anthers of indicated lines after Alexander staining. **(D)** Number of viable pollen per anther from flowers at positions 5–15 and 25–35 along the main inflorescence bolt. Error bars represent standard deviations (n = 9).

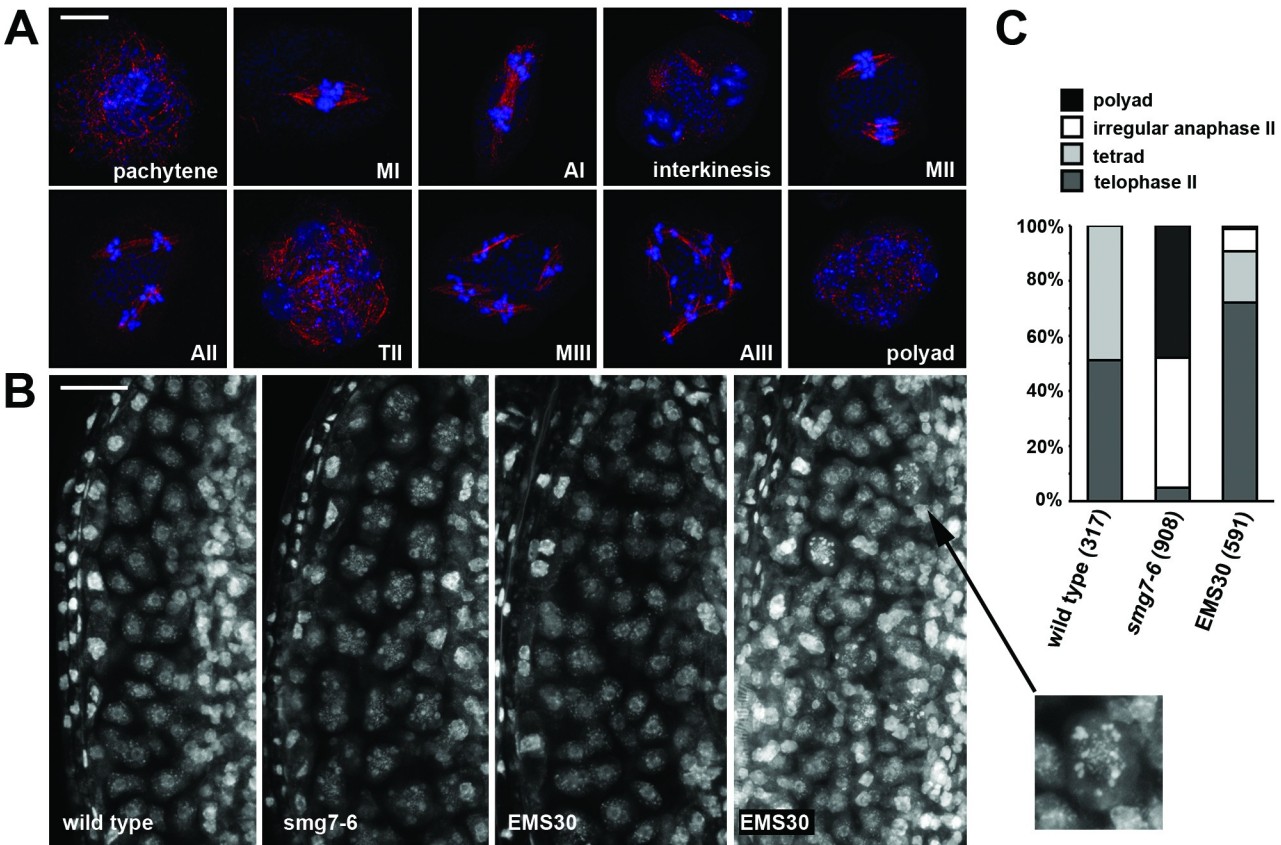

**Fig 5. Meiotic progression in the EMS30 line. (A)** Immunocytological analysis of EMS30 PMCs with spindles visualized with anti-α-tubulin antibody (red). DNA is counterstained with DAPI (blue). MI, MII, MIII—metaphase I, II, III; AI, AII, AIII—anaphase I,II, III; TII—telophase II. Scale bar represents 5 μm. **(B)** Anther loculi stained with DAPI. Wild type and *smg7-6* loculi contain tetrads and polyads, respectively. EMS30 plants contain tetrads (middle panel) and a mixture of polyads, tetrads, and AIII in their loculi (right panel). The inset shows a detail of a PMC in the MIII/AIII stage. Scale bar represents 20 μm. **(C)** Quantification of PMCs in late stages of meiosis. Number of PMCs analyzed for each genotype is indicated in parentheses.

genome using the ArtMAP mapping algorithm [34]. Although EMS30 and EMS155 contained different *de novo* mutations distributed throughout their genomes, they shared one identical polymorphism in the CENH3 gene at the end of the left arm of chromosome 1 (S5 Fig). The mutation was present with 100% frequency in the pools with DNA from fertile plants in the B2 populations derived from EMS30 and EMS155 lines. In both cases, the mutation represents a G to A transition in the splicing donor site of the 3rd exon of the CENH3 gene (Fig 6A). It is a silent mutation that does not alter amino acid sequence. Since this mutation was uncovered independently in two suppressor lines that do not share any other *de novo* polymorphism, we consider this mutation to be responsible for the restored fertility of *smg7-6* and refer to it as the *cenh3-4* allele.

By backcrossing the mutant suppressor line with wild type, selfing the progeny, and geno-typing in the F2, we also generated a line which carries the *cenh3-4* allele independent from the *smg7-6* mutation. As a single or double mutant, the *cenh3-4* mutation causes inefficient splicing and retention of the third intron, which results in an approximately 10-fold reduction of fully spliced *CENH3* mRNA (Fig 6B). We used two combinations of primers matching either wild type (CENH3-G) or mutant (CENH3-A) cDNA to ascertain that the observed reduction

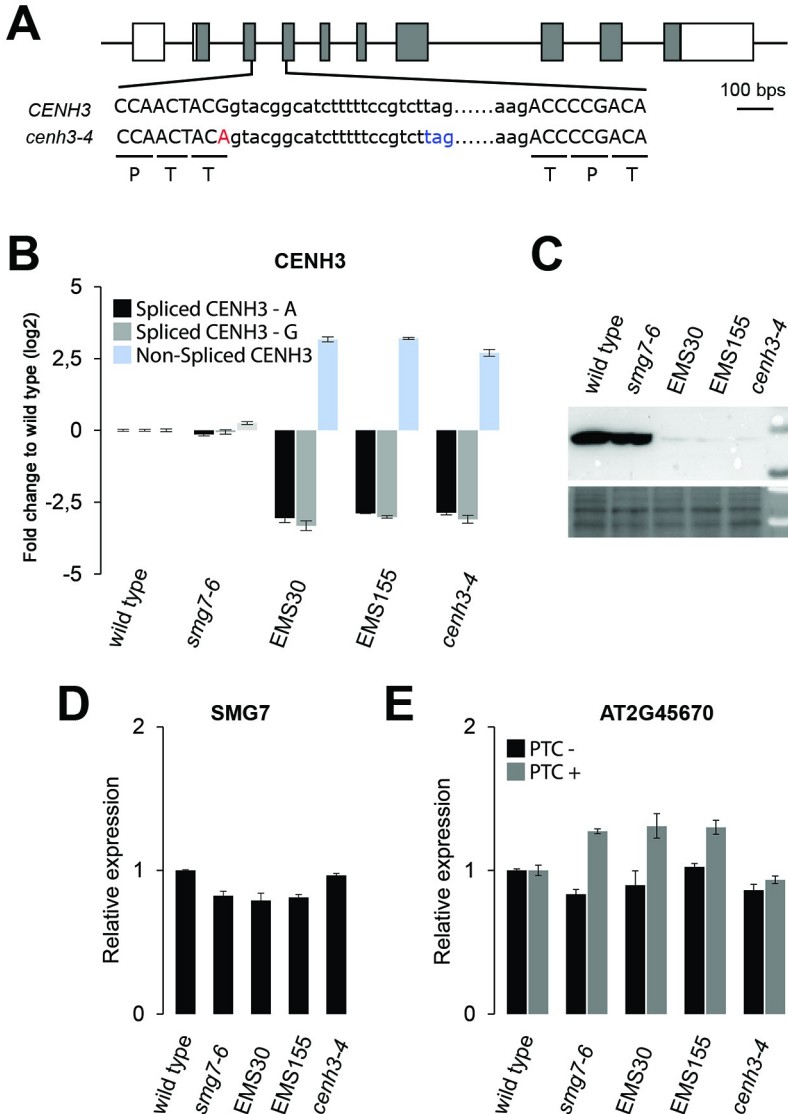

**Fig 6. Molecular characterization of the *cenh3-4* allele.** (**A**) Diagram of the CENH3 gene with exons marked as boxes. DNA sequence surrounding the intron 3 is shown; capital letters depict exons. Amino acids encoded by canonically spliced mRNA are indicated. The *cenh3-4* mutation at the splice donor site of exon 3 is indicated in red, stop codon in the intron of unspliced mRNA in blue. (**B**) Relative abundance of spliced and unspliced *CENH3* mRNA determined by quantitative RT-PCR. Two sets of primers were designed for fully spliced mRNA matching either wild type (CENH3-G) or *cenh3-4* (CENH3-A) allele sequences. (**C**) Western blot detection of total CENH3 protein by anti-CENH3 antibody. Protein loading is shown on a Ponceau-S stained membrane (bottom panel). (**D**) Abundance of SMG7 mRNA relative to wild type determined by qRT-PCR. (E) NMD efficiency assessed by qRT-PCR analysis of alternatively spliced variants of the AT2G45670 gene with or without a premature termination codon (PTC) relative to wild type. Error bars in (B), (D) ad (E) represent standard deviations from 3 biological replicas.

is due to lower level of spliced CENH3 mRNA and not because of inefficient RT-PCR amplification (Fig 6B). Since the mis-spliced mRNA encodes a short open reading frame comprising only the first 32 amino acids of CENH3 (Fig 6A), *cenh3-4* plants have a substantially decreased amount of CENH3 protein (Fig 6C). The *cenh3-4* mutation does not affect expression of *SMG7* mRNA or the efficiency of NMD, as the abundance of an endogenous transcript containing a premature termination codon is not altered (Fig 6D and 6E).

CENH3 is a histone H3 variant that is specifically localized to centromeres. In Arabidopsis, CENH3 is loaded onto centromeres during G2 and remains there throughout the cell cycle [35]. To determine whether CENH3 is present on centromeric chromatin in *cenh3-4* mutants, we performed immunolocalization using a CENH3 antibody. In wild type, we readily detected ten discrete dots in tapetum nuclei that likely correspond to ten centromeres, and approximately five pronounced dots in pachytene meiocytes that reflect synapsed centromeres of paired homologous chromosomes (Fig 7A and S6 Fig). In contrast, no signals were detected when we used the same imaging conditions in *cenh3-4* mutants (S6 Fig). Substantially increasing the exposure time revealed weak fluorescence at DAPI-dense regions, indicating limiting amounts of CENH3 at the centromeres (Fig 7A). A substantial reduction of the CENH3 was also observed on centromeres in root and leaf nuclei of *cenh3-4* mutants (S7 Fig). This was further validated by chromatin immunoprecipitation in seedlings that showed association of CENH3 with the Arabidopsis centromeric satellite repeat CEN180 in *cenh3-4* plants, albeit at a lower level compared to wild type (Fig 7B and 7C). Thus, the *cenh3-4* mutation substantially reduces the amount of CENH3 in centromeric chromatin.

CENH3 forms a hub for binding kinetochore proteins during chromosome segregation. Therefore, we next performed immunolocalization in *cenh3-4* nuclei for CENP-C and MIS12, key proteins of the inner kinetochore complex [36,37], and KNL2, a CENH3 assembly factor [38]. CENP-C, MIS12 and KNL2 signals in *cenh3-4* root nuclei were substantially lower than in wild type (Fig 8), suggesting that the decreased amount of CENH3 in the mutant affects the kinetochore structure.

## cenh3-4 affects mitotic progression and chromosome segregation

Considering the role of CENH3 in chromosome segregation not only during meiosis, but also during the mitotic cell cycle and progression through M-phase, it was striking that the somatic growth of *cenh3-4* plants was hardly affected, despite a drastically reduced level of CENH3. Mutant plants are viable, fertile, and do not show any gross growth retardation (Fig 4A). More thorough examination revealed that the roots of plants carrying the *cenh3-4* allele grow slower when cultivated on agar plates (Fig 9A and 9B), which is indicative of impaired cell division. To reveal whether mitosis is affected in *cenh3-4* plants, we performed live imaging of mitotic progression in root cells expressing the *HTA10*:*RFP* chromatin marker [33]. We determined the duration of early mitosis from nuclear envelope breakdown (NEB) through prometaphase to the end of metaphase, and of anaphase from chromatid separation to their arrival at the final position within the dividing cell (Fig 9C). We noticed that the duration of prometaphase/metaphase is approximately twice as long in *cenh3-4* mutants as in wild type, whereas anaphase is unaffected (Fig 9D, S4 and S5 Movies, Table 1).

A prolonged prometaphase indicates problems with chromosome biorientation and the formation of a bipolar spindle. Therefore, we next examined whether mitosis in *cenh3-4* mutants is sensitive to oryzalin, a dinitroaniline herbicide that disrupts polymerization of microtubules [39]. We found that treatment of roots with 1.5 μM oryzalin has a relatively mild effect on mitotic progression in wild type, causing extension of the prometaphase from 10 to 14 min, which was followed by a normal anaphase (Fig 9D, Table 1, S6 Movie). However, the same concentration of oryzalin inhibited chromosome biorientation and formation of a stable metaphase plate in *cenh3-4* mutants. All scored cells entering mitosis remained arrested in prometaphase and did not reach anaphase even after 1 hr of recording (Table 1, S7 Movie). The increased sensitivity to a microtubule inhibitor indicates that association of kinetochores with microtubules is impaired in *cenh3-4* plants.

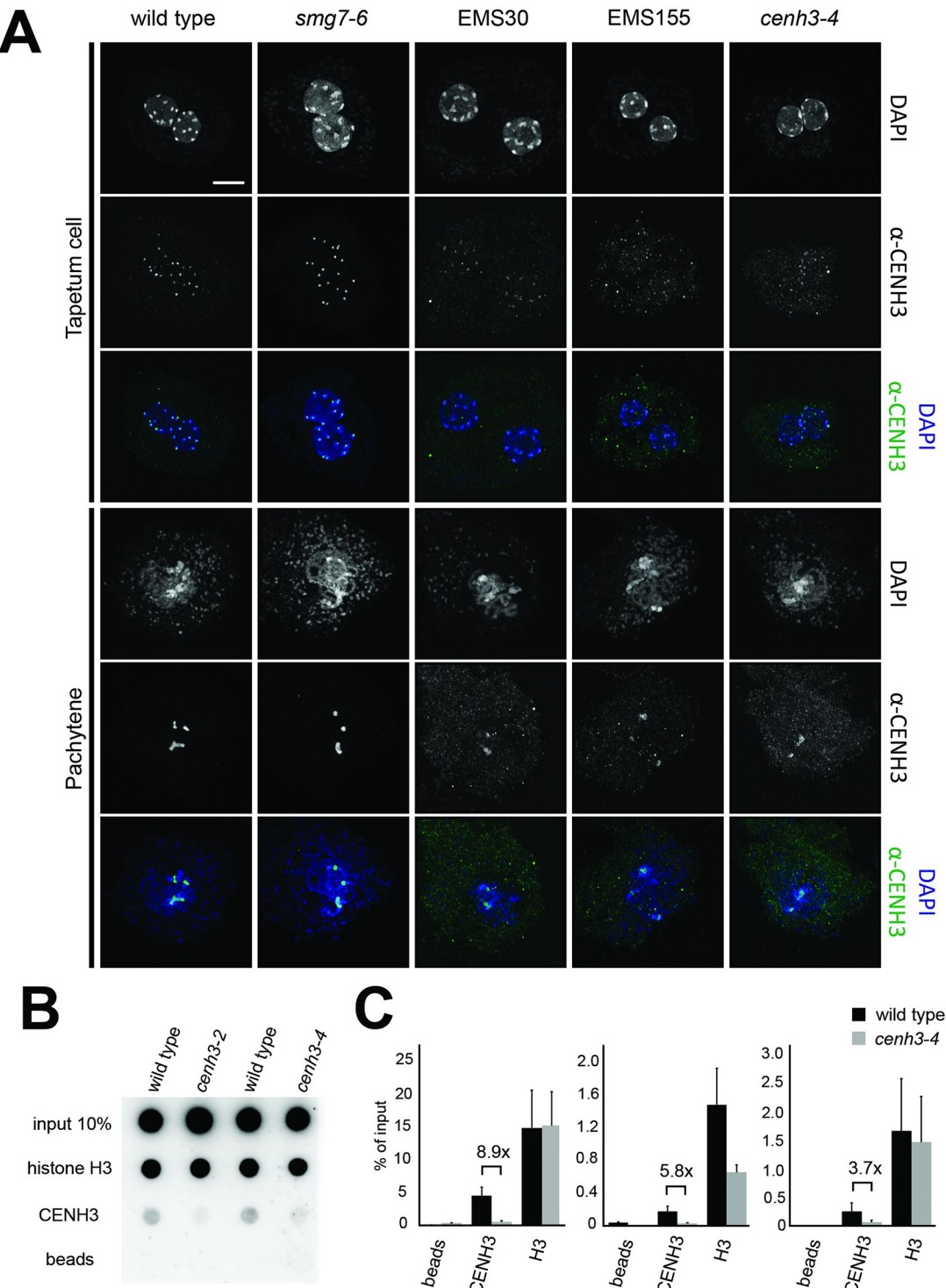

**Fig 7. Effect of *cenh3-4* mutation on the localization of CENH3 on Arabidopsis centromeres. (A)** Immunodetection of CENH3 in PMCs and tapetum cells using α-CENH3 antibody (green); DNA is counterstained with DAPI (blue). Longer exposure times for EMS30, EMS155, and *cenh3-4* were applied to detect the signal. Scale bar corresponds to 5 μm. **(B)** Association of CENH3 with the CEN180 satellite repeat determined by chromatin immunoprecipitation with CENH3 antibody and dot blot hybridization. Antibody against histone H3 was used as a control. **(C)** Quantification of the CEN180 repeat in the ChIP experiments by qPCR. Error bars represent standard deviation of three technical replicates. Three independent experiments are presented.

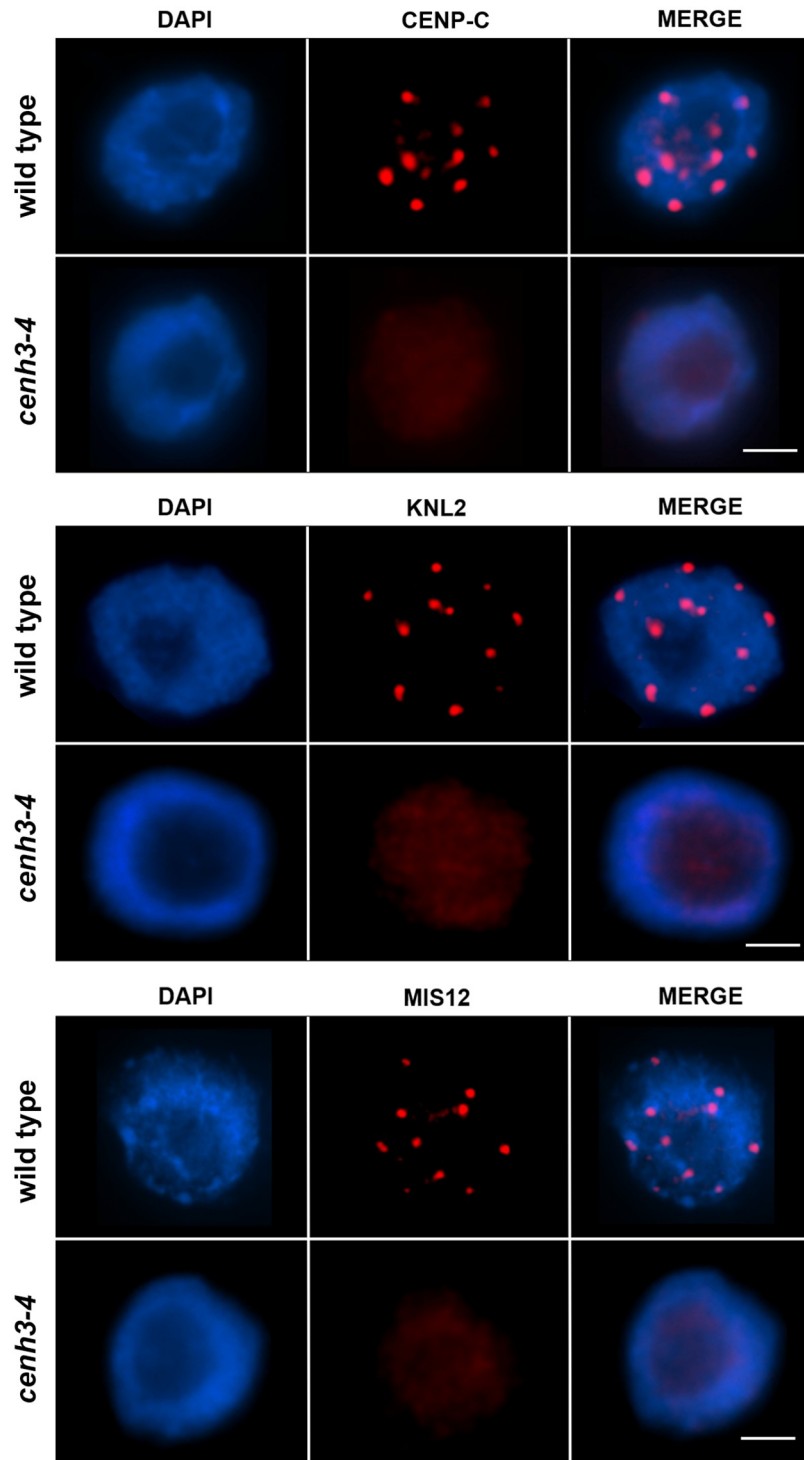

**Fig 8. Immunodetection of CENP-C, KNL2 and MIS12 in root nuclei of *cenh3-4* and wild type.** DNA is counter-stained with DAPI (blue). Longer exposure times for *cenh3-4* were applied to detect a signal. Scale bar = 5 μm.

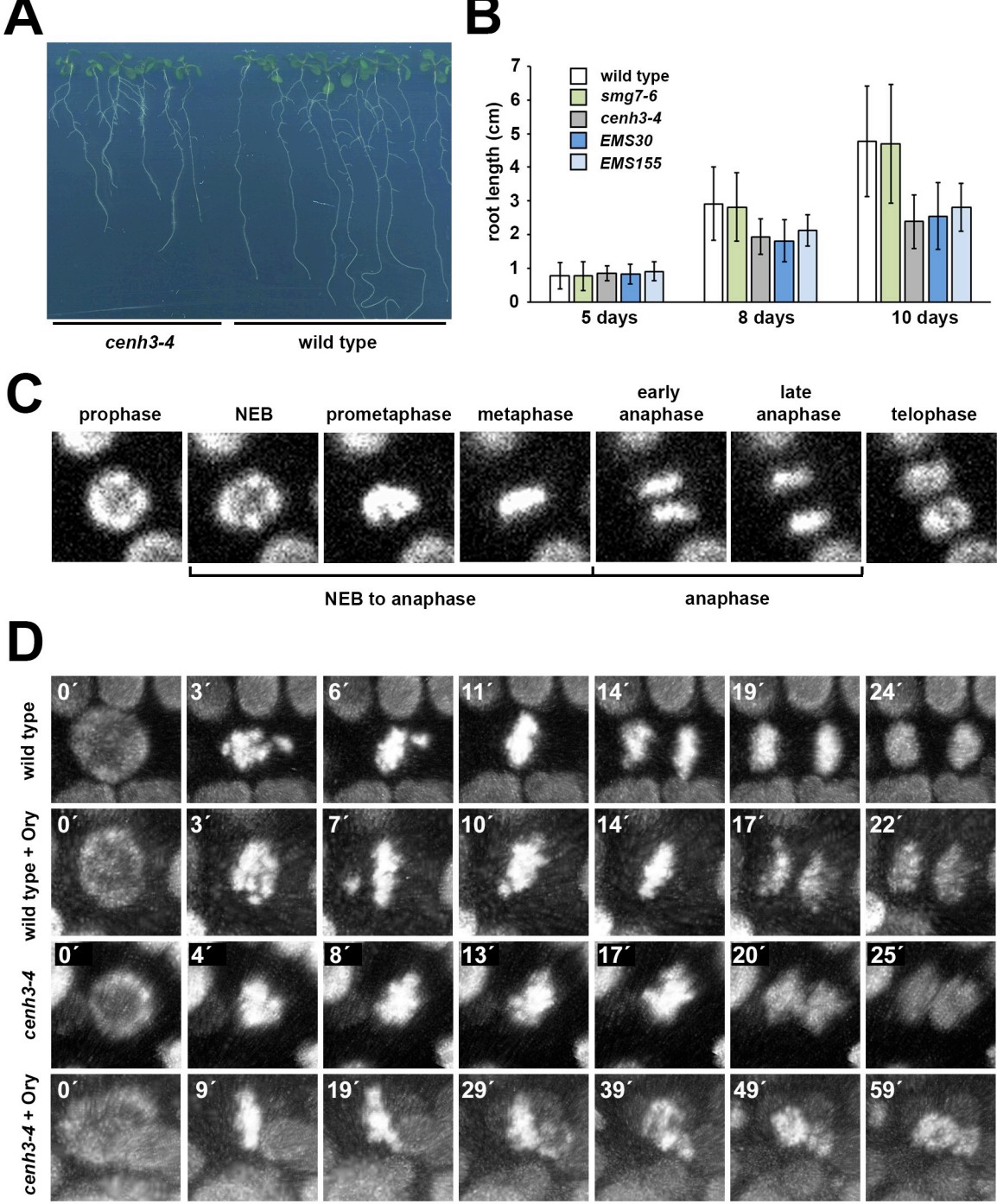

**Fig 9. Mitotic defects in *cenh3-4* mutants. (A)** Seedlings grown on vertically oriented agar plates for 10 d after germination. **(B)** Dynamics of root growth on vertical agar plate. Error bars represent standard deviations (n = 17 to 39). **(C)** An example of mitotic progression in a root cell visualized by the HTA10:RFP marker. Classification of stages whose duration was quantified in Table 1 is indicated. **(D)** Time-lapse series of mitotic nuclei from wild type and *cenh3-4* plants that were treated with 1.5 μM oryzalin (Ory) during the experiment.

**Table 1. Duration of mitotic phases in root cells.** NEB was defined as a time point when nuclei lost round shape (Fig 9C). None of the examined cells in *cenh3-4* roots treated with oryzalin that entered the mitosis formed a metaphase plate (n = 12).

| | | No oryzalin | | Oryzalin 1.5 μM | |
|---|---|---|---|---|---|
| Wild type | NEB to anaphase | 10.0 ± 1.3 | n = 9 | 13.8 ± 2.9 | n = 13 |
| | Anaphase | 4.2 ± 0.7 | | 4.2 ± 0.8 | |
| | No metaphase | n. a. | n = 0 | n. a. | n = 1 |
| *cenh3-4* | NEB to anaphase | 20.8 ± 5.9 | n = 14 | n. a. | n = 0 |
| | Anaphase | 4.2 ± 0.7 | | n. a. | |
| | No metaphase | n. a. | n = 0 | n. a. | n = 12 |

## The cenh3-4 allele does not promote the formation of haploid plants in crosses with wild type

Interference with CENH3 function through structural rearrangements in the N-terminal tail or amino acid substitutions in the conserved histone fold domain was shown to induce chromosome elimination and formation of haploid plants in Arabidopsis F1 crosses with wild type [40–43]. Some of the mutations were reported to decrease CENH3 levels, which suggested that downregulation of CENH3 could be sufficient to induce formation of haploid plants upon outcrossing [41,44]. To test this hypothesis, we pollinated *cenh3-4* mutants with pollen from trichomeless *gl1-1* plants. Haploid plants with this recessive mutant allele would be easily scored by their smooth leaf surface, in contrast to rough leaves in diploid hybrids. Out of 500 F1 plants, we identified only a single haploid plant validated by flow cytometry (S8 Fig). The frequency of haploid offspring (0.2%) in *cenh3-4* crosses is much lower than frequencies reported with mutations that alter the CENH3 amino acid sequence [40–43], suggesting that the decreased amount of CENH3 in *cenh3-4* mutants is insufficient to trigger uniparental genome elimination in Arabidopsis.

## Discussion

Transition from meiosis to post-meiotic differentiation of haploid gametophytes is governed by a dedicated mechanism that requires the nonsense-mediated RNA decay factor SMG7 [30]. So far, the only molecular function assigned to SMG7 was related to NMD, where SMG7 binds to phosphorylated UPF1 via its N-terminal phosphoserine binding domain and mediates re-localization of UPF1-bound RNA to P-bodies [45]. In vertebrates, the C-terminal region of SMG7 further associates with the CCR4-NOT deadenylase to degrade aberrant mRNA. This mechanism also appears to be conserved in plants [46,47]. Nonetheless, two lines of evidence argue that the meiotic function of SMG7 in Arabidopsis is not implemented through NMD. First, Arabidopsis UPF1-deficient plants are more impaired in NMD than SMG7-null mutants [48], but they do not show a meiotic phenotype and produce viable pollen [32]. Second, the hypomorphic *smg7-6* allele carrying a T-DNA insertion in the C-terminal region is NMD-proficient, but still exhibits meiotic defects [31,32]. We therefore conclude that SMG7 has an additional role in regulating meiotic exit, at least in the male pathway.

We previously proposed that SMG7 directly or indirectly contributes to the downregulation of CDK activity at the end of meiosis [29]. Processes required for chromosome segregation, such as nuclear envelope breakdown, chromosome condensation, and spindle formation are driven by increasing activity of M-phase CDKs. Degradation of M-phase cyclins by the APC/C in anaphase and downregulation of CDKs revert these processes and allow cytokinesis, exit from the M-phase, and licensing of origins of replication for DNA synthesis in the next S-phase [49,50]. If CDK activity is not irreversibly downregulated, untimely chromosome re-

condensation, spindle reassembly, and re-initiation of chromosome segregation might be consequences, as seen in *tdm1* and *smg7-6*. The complete lack of pollen in *tdm1* versus low numbers of viable pollen grains in *smg7-6* could originate from differences in the degree of re-condensation: less condensed chromatin in *tdm1* in the third division might preclude the formation of any chromosome cluster that would result in a functional microspore, while slightly stronger chromosome condensation at the respective state in *smg7-6* could increase the chance of encapsulating the right chromosome combination for the occasional formation of viable pollen.

This residual fertility in *smg7-6* permitted a suppressor screen to identify genes whose mutations affect meiotic exit in Arabidopsis and increase production of viable pollen and seeds. Identifying a mutation, twice independently, in the gene for centromeric histone CENH3 that partially rescues infertility of *smg7-6* plants came as a surprise. CENH3 is the key determinant of centromeric chromatin and kinetochore assembly. Its inactivation in Arabidopsis is embryonic lethal [40], and RNAi-mediated knock down of *CENH3* mRNA to 27–43% of the wild type level was reported to cause dwarfism and severe developmental defects [51]. Although the *cenh3-4* mutation leads to a 10-fold reduction of fully spliced *CENH3* mRNA and depletion of CENH3 from centromeres, mutant plants do not exhibit growth abnormalities under standard conditions and are fully fertile. Immunolocalization of CENH3 and the inner kinetochore proteins CENP-C and KNL2 suggest that centromeres and kinetochores are smaller in *cenh3-4* plants compared to wild type. These data indicate that plants can tolerate a substantial decrease of CENH3 level and centromere size. This has also been shown for centromeres on supernumerary B-chromosomes in maize that retained their functionality despite being trimmed to approximately 100 kb via chromosome fission [52].

How does the decreased level of CENH3 contribute to the restored fertility of *smg7-6* plants? The most notable cytological effect of the single *cenh3-4* mutant was delayed congression of mitotic chromosomes to the metaphase plate, suggesting less efficient establishment of a stable bipolar spindle. Chromosome congression and the bipolar spindle are formed by a search-and-capture mechanism in which microtubules initially establish lateral contacts with kinetochores that are later transformed into more stable attachments to the plus ends of microtubules [53,54]. The small kinetochores in *cenh3-4* mutants may decrease the efficiency of kinetochore-microtubule interaction and increase the time required for spindle formation. Such size-dependent attachment of kinetochores was predicted by computational modelling [54] and supported by the observation that chromosomes with larger kinetochores acquire bipolar orientation faster than chromosomes with smaller kinetochores [55]. Smaller kinetochores in the *cenh3-4 smg7-6* double mutant PMCs could therefore reduce the degree or speed of aberrant meiotic divisions compared to *smg7-6* single mutant plants. We propose that inefficient formation of centromere-microtubule interactions hinders spindle reassembly, re-entry into aberrant rounds of chromosome segregation, and thereby allows more efficient formation of viable pollen and higher fertility.

Centromere-mediated genome elimination is a promising strategy for inducing haploid plants for various breeding applications [56]. This technology was developed in Arabidopsis where all chromosomes of mutants with structurally altered CENH3 are eliminated upon crossing with wild type [40–42,57]. The underlying mechanism is assumed to be based on postzygotic incompatibility, whereby the parental chromosome set with the structurally altered CENH3 at its centromeres is mitotically unstable and therefore is left behind in early embryonic divisions [40,58,59]. It was hypothesized that CENH3 mutations may impair chromatin loading, forming smaller centromeres that cannot compete with the larger centromeres of the crossing parent [44], causing early loss of chromosomes due to size dimorphism of parental centromeres. Indeed, genome elimination can be induced in crosses between species with

centromeres of different size, and by a CENH3 mutation that affects centromere loading [41,44]. Furthermore, haploid plants were generated in crosses with maize heterozygous for a *cenh3* null mutation, suggesting that dilution of CENH3 during gametophytic divisions can render centromeres smaller or dysfunctional [60]. However, the substantially reduced level of CENH3 that, according to the immunocytological data, results in smaller centromeres and impaired mitotic spindles, was not efficient in haploid induction, at least in Arabidopsis *cenh3-4* mutants. Notably, a comparably low frequency of Arabidopsis haploids was found after reducing the amount of wild-type CENH3 through female gametogenesis in plants heterozygous for the *cenh3-1* null mutation [61]. Also in wheat, multiple knock-outs of homeologous CENH3 genes are insufficient to induce haploid plants unless combined with a hypomorphic mutation containing a short deletion in the CENH3 N-terminal domain [62]. Thus, the efficiency of centromere-mediated genome elimination may depend on the extent and combined effects of qualitative and quantitative changes in centromere structure.

## Methods

### Plant material and growth conditions

*Arabidopsis thaliana* ecotype Columbia (Col-0) and mutant seeds were grown on soil in growth chambers at 21˚C at 50–60% humidity under 16 h/8 h light/dark cycles. The following mutant lines were used in this study: *smg7-6* [32], *smg7-1*, *smg7-3* [30], *gl1-1* derived from *tert* line [63]. The *tdm1-4* mutant was obtained from NASC (SALK_123139) and PCR-genotyped using PCR primers described in S1 Table. Plants used for live cell imaging were generated by introgression of reporter constructs from *HTA10:RFP* [33] and *pRPS5A::TagRFP:TUB4* [22] lines. Root growth assay was performed by growing surface-sterilized seeds on vertically oriented MS agar plates (0.7% plant agar, Duchefa Biochemie) at 21˚C under 16 h/8 h light/dark photoperiods. The position of the root tip was marked 5, 8 and 10 days after germination to determine the root growth rate.

### Assessment of plant fertility

Pollen viability was determined by Alexander staining as described [64]. Silique length was measured when apical meristems ceased forming new flowers. Average silique length was calculated at each position for the first 40 siliques along the main inflorescence bolt (numbered with 1 for the oldest and 40 for the youngest silique).

### Genetic screening

Seeds from *smg7-6* plants were incubated in 50 ml of water at 4˚C overnight. Water was replaced with 50 ml of 0.3% (v/v) ethyl-methanesulfonate (EMS) in water and incubated for 8 h at room temperature in the dark with gentle shaking. The EMS solution was replaced with water and the seeds were incubated for 3 days at 4˚C in the dark. Twelve seeds were sown per pot (9x9 cm) and M2 seeds were pooled from all plants in one pot. Around 100 M2 seeds from each pool were grown and manually scored for improved fertility compared to *smg7-6* mutants grown in parallel. The genetic transmissibility of restored fertility was confirmed in the M3 generation and M3 plants were backcrossed to the parental *smg7-6* line to create B2 mapping populations. Inflorescences from segregants with improved fertility in B2 were pooled (at least 50 plants per line) for DNA extraction using the CTAB (cetyltrimethyl ammonium bromide) method [65]. DNA (2 μg) was sheared in a S220 Focused-ultrasonicator (Covaris), DNA fragments were purified using the DNA Clean &Concentrator-5 kit (Zymo Research), and quantified using the Quant-iT PicoGreen dsDNA Reagent (Thermo Fisher Scientific). After

analyzing the samples for proper fragmentation (~150 bp) in an Agilent 2100 Bioanalyser System (Agilent Technologies), DNA libraries were prepared using 300 ng of fragmented DNA following the instructions of the NEBNext Ultra II DNA Library Prep kit (New England Biolabs). Samples were sequenced on a HiSeq 2500 (Illumina) with the sequencing output of single ends with 100 nucleotides in size. Mutations associated with improved fertility were identified using ArtMAP software [34]. For further genetic experiments, the identified *cenh3-4* mutation was PCR-genotyped by the Derived Cleaved Amplified Polymorphic Sequences (dCAPS) method using primers described in S1 Table. The PCR product was cleaved with *Pst*I and separated in 2% (w/v) high-resolution agarose in TBE buffer. The amplicon of the wild type allele remained uncleaved with a size of 222 bp, while that of the *cenh3-4* allele was cut into 188 bp and 34 bp fragments.

## Cytology

Staining of PMCs in whole anthers was performed as previously described [66] with the following modifications: inflorescences were fixed in PEM buffer (50 mM Pipes pH 6.9, 5 mM EGTA pH 8.0, 5 mM MgSO4, 0.1% Triton X100) supplemented with 4% formaldehyde by 15 min vacuum infiltration and 45 min incubation at room temperature. Inflorescences were washed 3x with PEM buffer and buds smaller than 0.6 mm were dissected, transferred to 100 µl of PEM supplemented with DAPI (5 µg/ml), and stained for 1 h in the dark. Anthers were washed twice with 1 ml of PEM buffer for 5 min, then incubated in PEM buffer at 60˚C for 10 min and at 4˚C for 10 min. Anthers were washed once, mounted in Vectashield (Vector Laboratories), covered with cover slips, and examined on an LSM700 or LSM880 confocal microscope (Zeiss). Immunodetection of microtubules in pollen mother cells was performed using rat anti-α-tubulin antibody (Serotec) as previously described [29]. The same protocol was applied to detect CENH3 with a custom-made (LifeTein, https://www.lifetein.com) polyclonal antisera raised against the N-terminal peptide of CenH3 [67] and anti-Rabbit-Alexa Fluor 488 (ThermoFisher Scientific). KNL2, MIS12 and CENP-C were detected in root nuclei as previously described [68] using custom made anti-AtKNL2 (dilution 1:2000) [38], anti-MIS12 (1:1000) or anti-AtCENP-C (1:300) antibodies (http://www.eurogentec.com/) [51] and goat anti-rabbit rhodamine (Jackson Immuno Research Laboratories). DNA content in nuclei of haploid plants was determined by flow cytometry as previously described [69].

## Live cell imaging

Live cell imaging of spindles in PMCs was performed with the *pRPS5A*::*TagRFP*:*TUB4* marker [22] using a protocol developed for light sheet microscopy [33]. Briefly, the floral buds were embedded in 5MS (5% sucrose + ½ MS, Murashige & Skoog Medium including vitamins and MES buffer, Duchefa Biochemie) supplemented with 1% low gelling agarose (Sigma Aldrich) in a glass capillary (size 4, Brand). The capillary was mounted in the metal holder of the Light sheet Z.1 microscope (Zeiss), the agarose with the embedded floral bud was partially pushed out from the glass capillary into the microscopy chamber containing 5MS media, and imaged in Light sheet Z.1 microscope using 10x objective (Detection optics 10x/0.5), single illumination (Illumination Optics 10x/0.2), 561 nm laser (15% intensity) in 5min time increments. The image data were processed by ZEN Blue software (Zeiss). Live cell imaging of mitosis in roots was performed with the *HTA10:RFP* marker as follows: surface-sterilized seeds were germinated on 1 ml of ½ MS medium with 0.8% phyto-agar (Duchefa Biochemie) in petri dishes with a glass bottom (MatTek corporation). Once growing roots reached the glass bottom, they were imaged with an LSM780 inverted confocal microscope (Zeiss, 40x objective) in 1 min

intervals. Two ml of 2 µM oryzalin solution were added to plates 10 min before imaging. Images were processed using Zen Black (Zeiss).

## Chromatin immunoprecipitation

ChIP experiments were performed from 10 day old seedlings with chromatin sheared to approximately 500 bp fragments by sonication according to a previously described protocol [70]. Five µl of anti-histone H3 antibody (1 mg/ml; ab1791; Abcam), and 10 µl of anti-CENH3 antibody were used. Detection of centromeric 180 bp satellite DNA was performed by dot-blot hybridization as well as by qPCR. For dot-blot hybridization, 40 µl from 50 µl samples were combined with 10 µl of MILI-Q water and 6 µl of 3 M NaOH, incubated for 1 h at 65˚C and dot-blotted on Amersham HybondTM-XL membrane (GE Healthcare). DNA was fixed by UV using a UV crosslinker BLX-254 (Analytik Jena). After prehybridization at 65˚C for 2 h in hybridization buffer (7% SDS, 0.25 M sodium-phosphate buffer pH 7.2), membranes were hybridized at 65˚C overnight with denatured probes generated by Klenow-labeling of an Arabidopsis centromere 180 bp satellite fragment with $\alpha^{32}P$ dATP (DecaLabel DNA Labeling Kit, Thermo Scientific). The fragment was prepared by PCR amplification of Arabidopsis genomic DNA using the primer combination CEN-1 ATCAAGTCATATTCGACTCCA and CEN-2 CTCATGTGTATGATTGAGAT, followed by purification (NucleoSpin Gel and PCR Clean-up, Macherey-Nagel). Membranes were washed twice with 2x SSC, 0.1% SDS at RT for 5 min and twice with 0.2x SSC, 0.1% SDS at 65˚C for 15 min. Membranes were wrapped in Saran wrap and exposed to a phospho-screen which was scanned with a Typhoon FLA 7000 (GE Healthcare). Signals were quantified using ImageQuant software (GE Healthcare). For qPCR quantification, 1 µl from 50 µl samples was used in a 20 µl qPCR reaction containing 1x Light-Cycler 480 High Resolution Melting Master mix, 3 mM MgCl2, and 0.25 µM of each primer CEN-f CCGTATGAGTCTTTGGCTTTG and CEN-r TTGGTTAGTGTTTTGGAGTCG. Reactions were performed in technical triplicates and quantified as percent of input.

## Western blot analysis

For nuclei purification, 300 mg of inflorescences were collected in 15 ml Falcon tubes, frozen in liquid nitrogen, and homogenized with metal beads by vortexing. The disrupted tissue was resuspended in 5 ml of nuclei isolation buffer (NIB) (10 mM MES-KOH pH 5.3, 10 mM NaCl, 10 mM KCl, 250 mM sucrose, 2.5 mM EDTA, 2.5 mM ß-mercaptoethanol, 0.1 mM spermine, 0.1 mM spermidine, 0.3% Triton X-100), and filtered through two layers of Miracloth into a 50 ml Falcon tube. Nuclei were pelleted by centrifugation, resuspended in 1 ml NIB and collected again by centrifugation. The pellet was resuspended in 150 µl of N buffer (250 mM sucrose, 15 mM Tris-HCl pH 7.5, 60 mM KCl, 15 mM NaCl, 5 mM $MgCl_2$, 1 mM $CaCl_2$, 1 mM DTT, 10 mM ß-glycerophosphate, protease inhibitors). Nuclei were lysed by adding 40 µl 5x Laemmli loading buffer (Sigma) and boiling for 5 min. 40 µg of nuclear protein was separated by SDS-polyacrylamide gel electrophoresis. Separated proteins were transferred to PVDF membranes (Thermo Scientific) by electroblotting. The membranes were incubated in low-fat milk with rabbit anti-CENH3 antibody (1:5,000; ab72001; Abcam) for 12 h at 4˚C. Secondary anti-rabid-HPR conjugated antibody was diluted (1:5000) and incubated for 2 h. TBST (25 mM Tris-Cl, pH 7.5, 150 mM NaCl, 0.05% Tween-20, pH 7.5) was used to wash the membranes and the signal was detected using ECL Western Blotting Substrate (Pierce).

## RNA analysis

RNA was isolated from inflorescences using the RNeasy Plant Mini Kit (Qiagen). Samples were treated with TURBO DNA-free Kit (Ambion) to remove contaminants from genomic

DNA. cDNA was synthetized from 5 μg of RNA with the Maxima H Minus First Strand cDNA Synthesis Kit (Thermo Scientific) and oligo (dT)18 primer. cDNA was used as a template for quantitative PCR reactions using the FastStart Essential DNA Green Master (Roche) and transcript-specific primer pairs (S1 Table) on the LightCycler 96 System (Roche). The ΔΔCt method was used to calculate the relative quantification of transcripts [71]. MON1 (AT2G28390) was used as reference gene, and transcript levels for each genotype were normalized to wild type controls.

## Supporting information

**S1 Movie. Live imaging of *pRPS5A::TagRFP:TUB4* tubulin marker during the first and second meiotic division in wild a type plant.**
(MP4)

**S2 Movie. Live imaging of *pRPS5A::TagRFP:TUB4* tubulin marker during regular meiotic divisions and two postmeiotic spindle reassembly cycles in smg7-6 mutants.**
(MP4)

**S3 Movie. Live imaging of *pRPS5A::TagRFP:TUB4* tubulin marker from the 3rd to 6th cycle of spindle reassembly in *smg7-6* mutants.**
(MP4)

**S4 Movie. Live imaging of *HTA10:RFP* chromatin marker during mitosis in a root tip of a wild type plant.**
(AVI)

**S5 Movie. Live imaging of *HTA10:RFP* chromatin marker during mitosis in a root tip of a *cenh3-2* mutant.**
(AVI)

**S6 Movie. Live imaging of *HTA10:RFP* chromatin marker during mitosis in a oryzalin treated root tip of a wild type plant.**
(AVI)

**S7 Movie. Live imaging of *HTA10:RFP* chromatin marker during mitosis in a oryzalin treated root tip of a *cenh3-2* mutant.**
(AVI)

**S1 Fig. Allelic series of Arabidopsis *smg7* mutants. (A)** Five week-old Arabidopsis mutants homozygous for the indicated alleles. **(B)** Quantitative RT-PCR analysis of *SMG7* mRNA from the region located upstream of the T-DNA insertions and of transcripts targeted by NMD in *smg7-1* and *smg7-6* mutants. Two mRNA splice variants, one containing a premature termination codon (PTC+), were quantified for the *AT2G45670* locus. Error bars indicate standard deviations from three biological replicas.
(TIF)

**S2 Fig. Female fertility in *smg7-6* mutants. (A)** Siliques from wild type and smg7-6 plants produced by pollination with wild type pollen. Scale bar = 0.5 cm. **(B)** Box-plot diagrams showing quantification of seed count per silique from wild type (N = 14) and *smg7-6* plants pollinated with wild type pollen. Yield from siliques at positions 1 to 15 (N = 136) and 20 to 35 (N = 153) is indicated for *smg7-6* mutants. **(C)** Box-plot diagram showing quantification of seeds in siliques at indicating positions along the main inflorescence bolt with 1 indicating the lowest position. 5 to 14 siliques were counted per position.
(TIF)

**S3 Fig. Genetic complementation of *smg7-6* mutation with the endogenous *pSMG7*::*SMG7* gene construct.** Anthers from 10 independent T1 *smg7-6* transformants carrying the *pSMG7*::*SMG7* construct with viable pollen detected by Alexander staining. All T1 lines show a restoration of viable pollen.
(TIF)

**S4 Fig. Epistatic analysis of *smg7-6* and *tdm1-4* mutations. (A)** Anthers of the indicated mutants after Alexander staining. Viable pollen stain red. **(B)** Meiotic progression in PMCs of *smg7-6 tdm1-4* double mutants. Spindles are stained with anti-α-tubulin antibody (red), DNA is counterstained with DAPI. Scale bar corresponds to 5 μm.
(TIF)

**S5 Fig. Association mapping of *de novo* mutations with the *smg7-6* suppressor trait in EMS30 and EMS155 lines. (A)** Genome-wide distribution of *de novo* mutations and their frequency in fertile B2plants generated from backcrosses of EMS30 and EMS155 lines with parental *smg7-6* plants. **(B)** *De novo* mutations at the left arm of chromosome 1 in the EMS30 and EMS155 lines. Coordinates of these mutations in TAIR10 annotation are indicated.
(TIF)

**S6 Fig. Immunodetection of CENH3 in prophase I using the same imaging conditions.** CENH3 is visualized with CENH3 antibody; DNA is counterstained with DAPI (blue). Scale bar represents 5 μm.
(TIF)

**S7 Fig. Immunodetection of CENH3 in leaf (A) and root (B) nuclei.** CENH3 is visualized with CENH3 antibody; DNA is counterstained with DAPI (blue). Scale bar represents 5 μm.
(TIF)

**S8 Fig. Haploid plant obtained from the cross between *cenh3-2* and *gl1-1* Col-0. (A)** The haploid plant (indicated by arrowhead) was recognized based on its trichomeless phenotype. **(B)** Nuclear content of inflorescence nuclei from the haploid and a parent plant determined by flow cytometry.
(TIF)

**S1 Table. Primers used in this study.**
(XLSX)

**S2 Table. Numerical data underlying presented graphs.**
(XLSX)

## Acknowledgments

We thank to Arp Schnittger for providing the *pRPS5A*::*TagRFP*:*TUB4* line, Sona Valuchova for help with image processing, and Andreas Houben for helpful discussion. The genome sequencing was performed by the Next Generation Sequencing Facility at Vienna BioCenter Core Facilities (VBCF), member of the Vienna BioCenter (VBC), Austria. We also acknowledge the Plant Sciences Facility at Vienna BioCenter Core Facilities (VBCF), and the Plant Sciences Core Facility of CEITEC MU for support with plant cultivation. Microscopy was performed in the BioOptics facility at the IMP, and the core facility CELLIM of CEITEC.

## Author Contributions

**Conceptualization:** Claudio Capitao, Ortrun Mittelsten Scheid, Karel Riha.

**Formal analysis:** Claudio Capitao, Jaroslav Fulnecek.

**Funding acquisition:** Inna Lermontova, Ortrun Mittelsten Scheid, Karel Riha.

**Investigation:** Claudio Capitao, Sorin Tanasa, Jaroslav Fulnecek, Vivek K. Raxwal, Svetlana Akimcheva, Petra Bulankova, Lucie Crhak Khaitova, Manikandan Kalidass, Inna Lermontova.

**Methodology:** Pavlina Mikulkova.

**Project administration:** Ortrun Mittelsten Scheid, Karel Riha.

**Resources:** Inna Lermontova.

**Supervision:** Inna Lermontova, Ortrun Mittelsten Scheid, Karel Riha.

**Visualization:** Claudio Capitao, Sorin Tanasa, Jaroslav Fulnecek, Lucie Crhak Khaitova, Manikandan Kalidass, Karel Riha.

**Writing – original draft:** Karel Riha.

**Writing – review & editing:** Claudio Capitao, Inna Lermontova, Ortrun Mittelsten Scheid.

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
