## [Decision Letter · Decision Letter 0]

5 May 2021

Dear Prof. Riha,

Thank you very much for submitting your Research Article entitled 'A CENH3 mutation promotes meiotic exit and restores fertility in SMG7-deficient Arabidopsis' to PLOS Genetics.

The manuscript was fully evaluated at the editorial level and by independent peer reviewers. The reviewers appreciated the attention to an important problem and were in agreement that your paper is worthy of consideration, but they raised a number of questions and issues about the current manuscript.

Based on the reviews, we will not be able to accept this version of the manuscript, but we would be willing to review a much-revised version. We cannot, of course, promise publication at that time.

If you decide to revise the manuscript for further consideration at PLOS Genetics, please aim to resubmit within the next 60 days, unless it will take extra time to address the concerns of the reviewers, in which case we would appreciate an expected resubmission date by email to plosgenetics@plos.org.

[LINK]

Please do not hesitate to contact us if you have any concerns or questions.

Yours sincerely,

Ian Henderson

Associate Editor

PLOS Genetics

Claudia Köhler

Section Editor: Plant Genetics

PLOS Genetics

Reviewer's Responses to Questions

**Comments to the Authors:**

Reviewer #1: This manuscript describes a mutation in Arabidopsis that has extended mitotic dividions after meiosis in a leaky fashion. When a suppressor screen was performed, what was recovered was a hypomorphic allele of CENH3 due to problems with splicing. Immunostaining of this material showed a reduced amount of CENH3 at centromeres. It was postulated that this circumstance delayed mitotic spindle formation and thus allowed the formation of more tetrads at the end of meiosis than occurs in the original mutation. Tests of haploid induction resulted in only a single individual out of many, which could potentially just be the spontaneous rate in contrast to engineered copies of CENH3 that give higher rates of haploid induction. Others have postulated that haploid induction occurs because of smaller centromere size in the engineered versions but this case is not consistent with that view. The conclusions in the paper follow from the experiments.

Minor point:

Line 281. The chromosome mentioned in citation 51 does indeed have only 110 kb of canonical centromere sequences. This work was performed before the realization of the epigenetic nature of plant centromeres became known. Subsequent work on this chromosome (Liu et al 2015, PNAS 112: E1263-1271) found that it actually has a de novo centromere elsewhere and the canonical site is not active. The de novo centromere has a CENH3 domain of 288 kb, so that chromosome does indeed have a small centromere. Thus, the point of the authors is still correct. Because this is not a major consideration for the manuscript as a whole, the authors might consider citing instead Jin et al., 2005, Plant Cell 17: 1412-1423. The same chromosome was analyzed in that paper as well and that was also before the realization of epigenetic aspects of plant centromeres. However, a number of small centromeres were examined in this latter manuscript and the amount of CENH3 was semi quantified. Those centromeres all work albeit some have reduced transmission.

Reviewer #2: Capitao et al. describe the course of meiosis in the smg7-6 mutant (identified and initially described by them in their previous work). In this mutant, PMCs undergo additional rounds of chromosome condensation and spindle assembly, resulting in meiosis exit problems. The authors then use the smg7-6 mutant in the suppressor screen, using the infertility of the first 20 flowers as a phenotype. This screen resulted in finding two lines that turned out to carry an identical mutation in the CENH3 gene. This mutant allele (called cenh3-2) causes inefficient splicing and retention of the third intron, which translates into a 10-fold decrease in the amount of protein. The authors concluded that this in turn results in a reduction in binding of kinetochore proteins CENP-C and KNL2. The course of meiosis is slightly disturbed in the cenh3-2 mutant, in particular the prometaphase is prolonged, but the fertility (measured as pollen number per anther) remains practically unchanged. Apart from slow root growth and increased sensitivity to oryzalin, a microtubule inhibitor, no changes in the vegetative development of the mutant were observed, which are usually pronounced in lines with a reduced amount of CENH3. This allele was also unable to stimulate the formation of haploid plants, which is also a typical effect of CENH3 level reduction.

Overall it is an interesting work worth considering a publication in PLoS Genetics. The work broadens our understanding of the role of CENH3 and centromeres in chromosome segregation and also sheds some light on SMG7 function in meiosis. Elegant microscopic analyses, including experimental assays of the course of meiosis, are also worth noting. However, there are several elements that require further confirmation before the work is published.

Major concerns:

1. The suppressor screen used in the study is based on the infertility of the first 20-25 flowers (later flowers are partially fertile) in the smg7-6 mutant. However, the authors admit that pollen production shows a similar reduction in all flowers. The authors suggest that this is due to " unknown physiological factors that affect pollination or fertilization" (line 155). However, differences in the course of male and female meiosis in this mutant are also possible. Pollination influencing factors can come from the stigma or the pollen itself. Therefore, I suggest carrying out a simple experiment with the analysis of seed setting in siliques pollinated with pollen from wild plants and smg7-6 in the first 20 and later flowers.

2. It is not understandable for me why the cenh3-2 mutant does not show any more serious developmental disorders. Decreasing the CENH3 level to the level of 20-30% causes significant phenotypic effects (Lermontova et al. Plant J. 2011), while the authors indicate that the protein level in cenh3-2 corresponds to about 10% of the protein in wild type (and this is confirmed in RT-PCR of which corresponds to the level of the transcript). The authors should check the expression level more carefully by analysing it in vegetative tissues such as rosette leaves (at transcript and/or protein level).

3. In connection with the previous point, it seems necessary to check whether a reduction of CENH3 expression (e.g. in RNAi lines) will also suppress the smg7-6 phenotype. Alternatively, suppression can be checked for other weak CENH3 mutant lines. I assume this team has access to such mutants.

4. Analysis of CENP-C and KNL2 proteins is a nice extension of the proposed hypothesis, however microscope images can hardly be considered as quantitative. Authors should mitigate the conclusions unless they present alternative quantitative analysis of the levels of these proteins bound to centromeres in the cenh3-2 mutant.

5. In line 293 the authors present the hypothesis that smg7-6 phenotype suppression in cenh3-2 is caused by inefficient formation of centromere-microtubule interactions which hinder spindle reassembly. To confirm this hypothesis, it should be enough to reduce the amount of one of the kinetochore proteins, eg KNL2, and see if it also allows for suppression. Can the authors perform such an experiment?

Minor points:

Line 125: Please specify what genotype the description in this line refers to.

Line 182: Please specify: was the double suppressor carrying the two independent suppressor mutations? Please rephrase.

Fig. 6: Could you please add colours, especially to fig. B, so the figure is more readable?

Reviewer #3: The paper by Capitao et al consists of two parts. In the first, the authors characterize the effect of a hypomorphic mutation at SMG7. In the second part, they isolate and characterize a smg7-6 suppressor that results in low expression of CENH3 and smaller chromosomes. The authors’ conclusion is that smaller centromeres reduce the probability of capturing spindle fibers, thus lengthening the time to bipolar spindle attachment and delaying and hindering the deleterious additional divisions. This is a plausible explanation. Overall, the characterization of mutant and suppressor are done and explained competently. The isolation and characterization of the suppressor is the more interesting part of this paper. The data demonstrate that a ~90% loss of CENH3 expression reduces the centromere size by several folds, but has negligible effects on the growth phenotype of arabidopsis under standard conditions. Conveniently, this reduction has a very weak haploid inducer phenotype and may be inconsistent with the hypothesis that a difference in centromere size will result in haploid induction after hybridization (however, see below). The interaction of cenh3-2 with a spindle function mutation is also very interesting.

In conclusion, the topic, the specific findings, and their analysis have both the quality and impact to make this manuscript interesting to a broad audience. Some fixes and improvements are needed.

Criticism

Major points

The smg7-6 phenotype is complex and incompletely characterized. This is not a mortal flaw, but it complicates the analysis of the reduced CENH3 effect. By the way, the reported 1:3 live:dead ratio is unexpected. The figure seems to suggest that this is a frequent event. It needs statistical quantification: are there ever 2:2 events? The 1 viable: 3 dead pollen trait in the tetrad could be due to one spore capturing 5+n chromosomes and depleting the others to be 5-n. Alternatively, the live spore could be the only one that has 5 different chromosomes. Does the progeny display aneuploidy?

It makes sense to number the cenh3 mutants progressively with publication. Kuppu (2015) described a cenh3 mutant (A86V), although they did not number it. The cited biorXiv paper by Marimuthu [60] describes a knock-out mutation in CENH3 and calls it 3-3. This should make the mutation described in this manuscript cenh3-4, or alternatively 3-3 if that is 3.-4.

L212. The reduced root growth trait could be caused by independent linked mutations. Has the trait been confirmed for both independent instances of the cenh3-2 allele, or for the transheterozygote (cross of two mutants)? Alternatively, it should be complemented by transformation with a wild-type CENH3.

The delayed timing explanation for suppression of smg7-6 by cenh3-2 could be better supported if analyzed as shown in Fig.3B,C.

I wonder if another explanation for the suppression of smg7-6 is that cenh3-2 centromeres capture fewer spindle fibers and are less likely to be missegregated to a wrong pole or cell.

Fig.6. I found the figure unclear. In A, the splice products of wild type and mutant should be described. Have the cDNA’s been sequenced and is the structure of the resulting CENH3 the same as in the wild type? In B, the log relative expression is useful when comparing regulation, but in this case it is not clear, particularly because the text in Results uses a fold estimate. Why not use % mRNA standardized on WT? Why are both mutant and wild-type spliced forms measured in B? Spliced-CENH3-A. Why is a log used in B and not in D and E?

Figure 7 and Sup. FigS5. It would be helpful to quantify CENH3 the signals (Fig7 and Sup. FigS5)

Figure 8: The KNL2 signals do not show 10 spots as expected in the wild-type. Why? For CENP-C, the cenh3-2 mutant there are more than 10? Why?

Discussion: Some may agree with the author’s conclusion that the results provide no support for the Dawe model of centromere size causing haploid induction. However, others may instead conclude that this paper supports such a model because haploid induction was demonstrated. I suppose “it depends” on the definition of haploid induction. To some, 0.2% is nearly background level. To others, it may be a significant signal. So, what is the background level? Is the control result significantly different from the reported rate?

Minor points

Abstract: the statement of no haploids is contradicted by the data in Results

L103: ...mutants, which contain a T-DNA…. domain, grow normally…

L107 and other places: accepted usage is “wild-type” as single adjective (the wild-type strain), “wild type” as adjective + noun (the wild type grows tall)

L136: metaphase I – telophase II, interkinesis, and metaphase II telophase II in wild type are measured as 46, 38, and 39 min: was this meant to be “metaphase I -to telophase I” or telophase II? If the latter applies, the timing does not make sense.

L217: Prometaphase/metaphase duration in cenh3-2 is twice as WT yet the plant inflorescence displays normal growth: this needs an explanation.

Table 1. Please define NEB. The legend states that “none of the examined cells….that entered the metaphase formed a metaphase plate”. How would one know that a cell entered metaphase if a metaphase plate was not formed?

Fig.2B. Comparison to wild-type pictures should be added.

Fig.4. For clarity, it would be better to label the EMS30 and 155 lines with the genotype (smg7-6, cenh3-2). Figure 4B. It shows silique length, but no data on the female fertility (this is in the context of cenh3-2, mutant characterization).

Fig.6. There is no doubt that this mutation results in a decreased amount of CENH3. It is not clear, however, how much lower. Is it 20, 10 or 5% of the wild-type amount? The western seems to indicate a quantity of CENH3 lower than that found in the ChIP. The assay and tissue is of course different, but one wonders what is the concentration.

Fig.7. A normal young cell in the anther maternal tissue would be more informative than the tapetum, which is typically endopolyploid. Some of the panels in Fig.S5 would be useful to convey the effect of longer exposure time. Panel B did not show up in my download. I saw what could be the same panel in the authors biorXiv preprint.

Fig.7. 180bp FISH coupled with CENH3 immuno (immuno-FISH) may be more complete.

Discussion: Worth mentioning: the centromere reduction may have a distinct deleterious effect on fitness in nature.

Method: did the ChIP analysis include a nuclease treatment? In other words, was it based on nucleosomes? This is an important detail for quantification and should be indicated in the method in addition to the reference [69].

**Have all data underlying the figures and results presented in the manuscript been provided?**

Reviewer #1: Yes

Reviewer #2: None

Reviewer #3: Yes

PLOS authors have the option to publish the peer review history of their article (what does this mean?). If published, this will include your full peer review and any attached files.

Reviewer #1: No

Reviewer #2: No

Reviewer #3: No

---

## [Decision Letter · Decision Letter 1]

16 Aug 2021

Dear Dr Riha,

We are pleased to inform you that your manuscript entitled "A CENH3 mutation promotes meiotic exit and restores fertility in SMG7-deficient Arabidopsis" has been editorially accepted for publication in PLOS Genetics. Congratulations!

Yours sincerely,

Gregory P. Copenhaver

Editor-in-Chief

PLOS Genetics

Claudia Köhler

Section Editor: Plant Genetics

PLOS Genetics

Comments from the reviewers (if applicable):

Reviewer's Responses to Questions

**Comments to the Authors:**

Reviewer #1: I am satisfied with the revision.

Reviewer #2: The authors provided additional experiments to answer my and other reviewers’ comments. The manuscript has been significantly improved in comparison to its original version. I have no further suggestions.

Reviewer #3: I am satisfied with the authors' response to my criticism and to that of the other reviewers. I wish to add my opinion to the issue of CENH3 depletion and effects. Reviewer 2 raises an interesting question about the apparent discrepancy between Lermontova RNAi experiment and the effect of this mutation, and to the matter of what CENH3 depletion level is compatible with regular mitosis. I am not sure, however, whether adding RNAi experiments to this work would address the matter. RNAi experiments, in my opinion, while they can be very useful, are more difficult to interpret than genetic mutations.

**Have all data underlying the figures and results presented in the manuscript been provided?**

Reviewer #1: Yes

Reviewer #2: Yes

Reviewer #3: Yes

PLOS authors have the option to publish the peer review history of their article (what does this mean?). If published, this will include your full peer review and any attached files.

Reviewer #1: No

Reviewer #2: No

Reviewer #3: No

**Data Deposition**

http://datadryad.org/submit?journalID=pgenetics&manu=PGENETICS-D-21-00504R1

**Press Queries**

---

## [Editor Report · Acceptance letter]

21 Sep 2021

PGENETICS-D-21-00504R1 

A CENH3 mutation promotes meiotic exit and restores fertility in SMG7-deficient Arabidopsis 

Dear Dr Riha, 

We are pleased to inform you that your manuscript entitled "A CENH3 mutation promotes meiotic exit and restores fertility in SMG7-deficient Arabidopsis" has been formally accepted for publication in PLOS Genetics! Your manuscript is now with our production department and you will be notified of the publication date in due course.

With kind regards,

Agnes Pap

PLOS Genetics

On behalf of:
